# Genetic Diversity and Population Divergence of a Rare, Endemic Grass (*Elymus breviaristatus*) in the Southeastern Qinghai-Tibetan Plateau

**Qingqing Yu** [1] , **Qian Liu** [2], **Yi Xiong** [1], **Yanli Xiong** [1], **Zhixiao Dong** [1], **Jian Yang** [1], **Wei Liu** [1], **Xiao Ma** [1,*] **and Shiqie Bai** [3,*]

[1]  College of Animal Science and Technology, Sichuan Agricultural University, Chengdu 611130, China; yuqinggzu93@126.com (Q.Y.); xiongyi95@126.com (Y.X.); yanlimaster@126.com (Y.X.); dongzhixiao@126.com (Z.D.); jianyang9999@126.com (J.Y.); lwgrass@126.com (W.L.)
[2]  Institute of Animal Husbandry and Veterinary Science of Liangshan Prefecture, Xichang 615024, China; liuqianxichang@aliyun.com
[3]  Sichuan Academy of Grassland Science, Chengdu 61110, China
*  Correspondence: maroar@126.com (X.M.); baiforage@163.com (S.B.); Tel./Fax: +86-028-86291010 (X.M.)

**Abstract:** *Elymus breviaristatus* is a grass species only distributed in the southeast of Qinghai-Tibetan Plateau (QTP), which has suffered from serious habitat fragmentation. Therefore, understanding patterns of genetic diversity within and among natural *E. breviaristatus* populations could provide insight for future conservation strategies. In this study, sequence-related amplified polymorphism markers were employed to investigate the genetic diversity and hierarchical structure of seven *E. breviaristatus* populations from QTP, China. Multiple measures of genetic diversity indicated that there is low to moderate genetic variation within *E. breviaristatus* populations, consistent with its presumed mating system. In spite of its rarity, *E. breviaristatus* presented high genetic diversity that was equivalent to or even higher than that of widespread species. Bayesian clustering approaches, along with clustering analysis and principal coordinate analysis partitioned the studied populations of *E. breviaristatus* into five genetic clusters. Differentiation coefficients ($F_{st}$, $G_{ST}$, etc.) and AMOVA analysis revealed considerable genetic divergence among different populations. BARRIER analyses indicated that there were two potential barriers to gene flow among the *E. breviaristatus* populations. Despite these patterns of differentiation, genetic distances between populations were independent of geographic distances (r = 0.2197, *p* = 0.2534), indicating little isolation by distance. Moreover, despite detecting a common outlier by two methods, bioclimatic factors (altitude, annual mean temperature, and annual mean precipitation) were not related to diversity parameters, indicating little evidence for isolation caused by the environment. These patterns of diversity within and between populations are used to propose a conservation strategy for *E. breviaristatus*.

**Keywords:** *Elymus breviaristatus*; SRAP; genetic diversity; outlier; population structure; isolation by distance (IBD); conservation strategy

---

## 1. Introduction

Genetic diversity within a taxon is considered to be critical for maintaining the long-term survival and continued evolutionary potential of populations or species [1]. Habitat modification and fragmentation as a consequence of human activities and global environmental change are serious problems affecting the preservation of biodiversity in many terrestrial ecosystems and a clear threat to survival of many species. It could negatively affect the spatial arrangement of habitat patches and further weaken the connectivity between populations [2–4]. Thus, habitat fragmentation can ultimately

lead to inbreeding, diminished evolutionary potential, and a high risk of extinction [5], particularly for rare and narrowly endemic plants. Accurate estimates of genetic diversity and population structure are therefore needed to provide insight into the population dynamics, adaptation, and evolution of species. In addition, genetic data will assist in the development of effective conservation strategies and sustainable utilization of germplasm resources [6]. Although patterns of genetic diversity and spatial genetic structure are affected by levels of dispersal, endemic rare species will be more likely influenced by heterogeneity in environmental and geographical factors that will limit dispersal.

On the eastern most fringe of the Qinghai-Tibetan Plateau (QTP), the northwestern Sichuan Plateau (NSP, Figure 1) is a global biodiversity hotspot in the Hengduan Mountains and one of the largest pastoral regions in China [7]. Low massifs and widespread alpine meadows make up its complex and various landscape, with an average elevation of more than 3500 m. In addition, the habitats of NSP are characterized by harsh environmental conditions, such as low temperature, low precipitation, strong winds, intense ultraviolet radiation, thin oxygen, and barren soil [1], which make the local ecosystem vulnerable and sensitive to habitat destruction and climate changes. In addition, this region harbors exceptional species richness and a high level of endemism, probably due to its unique geographic location, complex terrains, and the recent uplift of the QTP [1]. However, many endemic plant species have recently become endangered or threatened because of increased human disturbance and consequent habitat deterioration of rangelands [8].

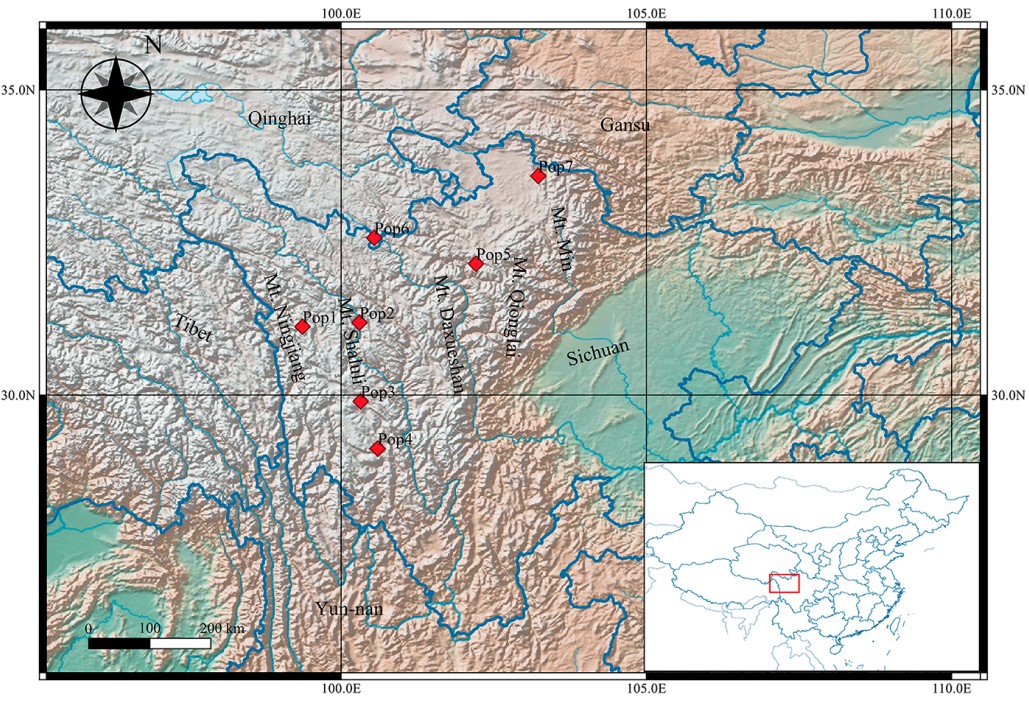

**Figure 1.** Locations of the *E. breviaristatus* populations used in this study.

The genus *Elymus*, which contains roughly 150 perennial species, is the largest and exclusive allopolyploid genus in the grass tribe Triticeae. It is found from the Arctic to temperate and subtropical regions [9]. The QTP and Xinjiang province of China is the current center of diversity of the genus, and it contains the largest number of indigenously congeneric species that span various ecological niches, diverse morphological characters, and alloploidy levels and strong resistance to biotic and abiotic stresses [10]. *Elymus breviaristatus*, an allohexaploid with the StHY genome (2n = 6x = 42) [11], is a bushy, self-pollinated, short-lived perennial grass species endemic to the QTP. It is mainly distributed in the NSP and to the south of the Qinghai Province at altitudes ranging from 3400 to 3600 m. It can be found as an occasional species in alpine meadows and shrubs [12]. This plant is strongly tolerant to

drought and cold and occurs in neutral or slightly alkaline sandy soils. It can grow well under artificial cultivation in regions of 4100 m. *E. breviaristatus* has a soft texture, high tillering ability, and high levels of crude proteins and is favored by livestock. Furthermore, it is worth noting that this species is characterized by a shorter lemma awn length (2–5 mm) than other congeneric species [13], which could greatly reduce the cost of mechanized harvesting and sowing of seeds. Consequently, it could be utilized in restoration and improvement of natural pastures or for the establishment of cultivated grasslands [14]. Because of the deterioration or fragmentation of natural habitats from global climate change and/or anthropogenic impacts, like livestock overgrazing in high altitude pastures, the wild resources of this species are declining [12]. In fact, it is now considered to be vulnerable in China and has been added to the important wild conservative plants list (Class II) in China [12]. In addition, natural populations of *E. breviaristatus* in the NSP are isolated by several local mountain ranges and rivers (such as the Shaluli, Daxueshan, and Qionglai mountains and Yalong and Daduhe rivers), which has further intensified fragmentation.

Despite previous work in *E. breviaristatus* demonstrating its tolerance to abiotic stress, performance of growth and herbage yield [15], phenotypic diversity [13], breeding and domestication of wild germplasm [16], and phylogenetic relationships [17], little attention to date has been paid to the population structure across its distribution in the QTP. Only two previous reports using isozyme and SSR marker analysis have been published, both of which found higher genetic variation among populations and moderate variation within populations [18]. Because of the limited number of markers and populations used, this previous study focused on population demography rather than adaptive evolution driven by environmental factors. Thus, it remains necessary to analyze *E. breviaristatus*'s genetic diversity, population structure, and any relationships with eco-environmental factors.

In recent decades, numerous types of genetic markers have been utilized in plant conservation biology to offer detailed information about genetic diversity and structure of fragmented natural populations, especially for rare and/or narrowly endemic plant species [1]. In contrast with other genetic markers, DNA-based markers are highly reproducible and efficient for estimating genetic variability. Here, we use a recently developed dominant molecular marker, known as sequence-related amplified polymorphisms (SRAPs). SRAPs are PCR-based markers that were developed to specifically amplify coding regions of the genome using nonspecific primers targeting GC-rich exons and AT-rich promoters, introns, and spacers [19]. The method is convenient, low-cost, and effective for producing genome-wide fragments with high reproducibility and versatility, it exhibits high co-dominance, and does not require *a priori* genomic information [19]. Therefore, this marker is suitable for conservation genetics investigation. Moreover, this technique has been widely used in studies of plant population genetics, especially for endangered species [20] or other congeneric species of *Elymus*, e.g., *E. nutans* [21] and *E. sibiricus* [22].

The genetic composition of seven natural populations of *E. breviaristatus*, across their main distribution along the NSP in China, were investigated by using SRAP markers. The main objectives were to (1) reveal the genetic diversity and population structure of natural populations of *E. breviaristatus*; (2) detect outlier loci and the status of linkage disequilibrium of SRAP markers in *E. breviaristatus*; (3) explore the association between genetic variation and eco-geographical factors or geographical distance. This genetic information could provide insights into the effective conservation of this species in the future.

## 2. Materials and Methods

### 2.1. Plant Sampling

For this study seed material of a total of 103 individuals of *E. breviaristatus* from seven populations of different sizes were collected across highly heterogeneous landscapes of the northwestern Sichuan Plateau (NSP) (Figure 1) in autumn 2010. At each location (population), individual spikes were collected by individual maternal parent plants. Wherever possible, a minimum distance of 5 m between

sampled plants was separated so as to minimize relatedness within a local population. Geographical distances between studied populations ranged from approximately 90 km (Pop1 and Pop2, Pop3 and 4) to over 1100 km (Pop4 and 5). A single seed from each collected singled spike was then planted into individual pots containing soil mixture, and maintained in a growth chamber with an approximate temperature 23 °C and cool white fluorescent lamps at 16 h photoperiod (illumination intensity 300 $\mu$mol·m$^2$·s$^{-1}$). Vouchers of the materials used are kept at the Sichuan Academy of Grassland Science, China. Table 1 recorded geographical coordinates, elevation, habitat, and three climatic variables for each population source, thereinto climatic variables were obtained from WorldClim—Global Climate Data (https://www.worldclim.org/) by DIVA-GIS 7.5 (http://www.diva-gis.org/). The populations were tentatively divided into three categories (high-H, medium-M, and low-L) based on ecological environment variation. The collection locations reflected the range of environments and climatic conditions over which *E. breviaristatus* occurs, ranging from 3110 to 3920 m in elevation, 0.8 to 6.7 °C in mean annual temperature, and 596 to 758 mm in annual precipitation.

## 2.2. DNA Extraction and PCR Amplification

Genomic DNA was extracted from leaves of young plants using the DP350 Plant DNA Kit (Tiangen Biotech., Beijing, China). A total of 19 primer pairs that amplified reproducibly and demonstrated polymorphism among individuals were selected from 220 SRAP primer pairs (Table S1). We set up 15 $\mu$L reactions containing 7.5 $\mu$L 2 × Taq PCR Mix (400 $\mu$M dNTP each, 15 mM Tris–HCl, 75 mM KCl, 2.0 mM MgCl$_2$) (Tiangen Biotech., Beijing, China), 300 nM of each primer, 30 ng total genomic DNA, and 1U Taq DNA polymerase. Cycling conditions on the Veriti96 Thermal Cycler (ABI, USA) were as follows: 5 min denaturation at 94 °C; five cycles of 94 °C for 1 min, 35 °C for 1 min, and 72 °C for 90 s; 35 cycles of 94 °C for 1 min, 51 °C for 1 min, and 72 °C for 90 s; and 10 min 72 °C for final extension. The amplified products were separated on 6% non-denaturing polyacrylamide gels and observed by silver nitrate staining.

## 2.3. Data Statistic and Analysis

Clearly amplified PCR bands were scored as 1 (presence) or 0 (absence) and then transformed into a binary matrix. In order to reduce the deviation caused by parameter overestimation in SRAP and other dominant marker data by 5%, we pruned any loci with band frequencies higher than 1-(3/N), where N is the number of individual samples, as proposed by Lynch and Milligan [23].

We used the POPGENE version 1.31 program to estimate genetic diversity, including the percentage of polymorphic bands (PPB), Nei's (1973) gene diversity (H$_e$), and Shannon's index (I). In addition, the expected heterozygosity (H$_j$) and Bayesian gene diversity (H$_B$), which were computed by AFLPsurv and HICKORY programs, respectively, were utilized to evaluate the genetic divergent of populations. We also used the polymorphic information content (PIC) to evaluate the discriminatory power of each SRAP primer.

Linkage disequilibrium (LD) among marker pairs was estimated using the squared allele-frequency correlation (r$^2$) for pairs of loci using Tassel 2.1. On the basis of summary statistics under Wright's [24] infinite island migration-drift model at equilibrium, the Dfdist method of Arlequin version 3.1 program [25] and the Bayesian method of BayeScan software [26] were used to detect outlier loci among the seven populations in *E. breviaristatus*. We discarded all alleles whose frequency was either greater than 99% or less than 1%.

**Table 1.** Location and geographic characteristic of the *Elymus breviaristatus* populations.

| Pop No. | Collection Site | Latitude | Longitude | Habit and Its Area | Sample Size | Population Size | Altitude (m) | Annual Temperature (°C) | Annual Precipitation (mm) |
|---|---|---|---|---|---|---|---|---|---|
| Pop1 | Anzi Town in Baiyu County of Garze region | N 31°7′16″ | E 99°22′4″ | Hillside; about 320 m$^2$ | 13 | 47 | 3890(H) | 1.3(L) | 596(L) |
| Pop2 | Anle Town in Xinlong County of Garze region | N 31°11′16″ | E 100°18′5″ | river valley; about 180 m$^2$ | 15 | 31 | 3170(L) | 5.8(H) | 638(L) |
| Pop3 | Benge Town in Litang County of Garze region | N 29°53′48″ | E 100°19′16″ | shrub grassland; about 176 m$^2$ | 15 | 28 | 3920(H) | 3.7(M) | 710(H) |
| Pop4 | Labo Town in Litang County of Garze region | N 29°7′25″ | E 100°35′58″ | river valley; about 210 m$^2$ | 15 | 19 | 3450(M) | 6.6(H) | 693(M) |
| Pop5 | Dazang Town in Maerkang County of Aba region | N 32°9′6″ | E 102°12′38″ | shrub grassland; about 136 m$^2$ | 15 | 23 | 3110(L) | 6.7(H) | 758(H) |
| Pop6 | Nianlong Town in Seda County of Garze region | N 32°34′36″ | E 100°32′31″ | shrub grassland; about 142 m$^2$ | 15 | 21 | 3680(M) | 0.8(L) | 662(M) |
| Pop7 | Baxi Town in Ruoergai County of Aba region | N33°35′21″ | E 103°13′46″ | shrub grassland; about 158 m$^2$ | 15 | 18 | 3130(L) | 3.4(M) | 654(M) |

Note: Different eco-geographic factors divided the populations into three groups. Based each size of the eco-geographic factors, the 'L' in the table represents the low level; 'M' represents the medium level; 'H' represents the high level.

We also calculated genetic differentiation among populations using $G_{ST}$ and $F_{st}$ in POPGENE and AFLPsurv version 1.0 software. In addition, POPGENE was used to calculate gene flow (Nm) between different geo-groups. The variance within and among populations was determined using analysis of molecular variance (AMOVA) in GenAlEx 6.51b2 project [27], with the Garze and Aba as groups. Other groups also were tested and were defined on the basis of geographical factors (altitude, AMT (Annual temperature) and AMP (Annual precipitation)). We also used AMOVA to estimate $\Phi_{ST}$, an analog of $F_{st}$ [27], followed by the permutation procedure (9999 replicates) for significance testing of variance components.

Furthermore, a Bayesian clustering method was utilized to identify the genetic structure of the *E. breviaristatus* populations using STRUCTURE version 2.3 program [28]. This analysis was performed under an admixture model that assumed independent allele frequencies and used 10,000 burn-in cycles followed by 100,000 Markov chain Monte Carlo iterations [28]. The batch run function performed a total of 70 runs (10 runs each for 1–7 clusters). The best value of the number of clusters (K) was determined for the *E. breviaristatus* populations using the maximum value of Ln K (K = 2–6) and the modal value of ΔK (K = 1–7) [29]. Then, the 10 replicates were clustered and aligned using CLUMPP 1.1 [30]. We then used Dice's coefficients to calculate a matrix of genetic similarity [31]. In addition, we calculated an individual-based principal coordinate analysis (PCoA) and an individual-based dendrogram with the unweighted pair-group method of averages (UPGMA) using the NTSYS-pc version 2.1, allowing generation of graphical representations of the genetic distance among the individuals. Nei's (1978) unbiased genetic distance among populations was calculated by AFLPsurv version 1.0 program, and then a population-based dendrogram was constructed with 9999 permutations bootstrapping to evaluate the robustness.

Finally, Monmonier's maximum difference algorithm in BARRIER version 2.2, which is based on genetic distance matrices, was used to understand the geographical location associated with the main genetic barriers among populations. We then used a Mantel test (1967) in NTSYS-pc version 2.1 software to estimate the correlation between Nei's (1978) genetic distance and geographical distance, or eco-geographic factors, respectively. We then calculated the Spearman correlation to estimate the correlation between the population diversity parameters ($H_e$, $H_o$, $H_B$, and $H_j$) and climatic factors (altitude, AMT, and AMP).

## 3. Results

### 3.1. SRAP-PCR Amplification and Outlier Analysis

In this study, 19 appropriate primer combinations (PCs) were screened out of a total 220 PCs. These selected PCs produced stable, repeatable amplification patterns in duplicate experiments, which produced a total of 459 identical and well-amplified bands ranging in length from 200 to 2000 bp. The number of total bands produced by each PC varied from 20 (Me2-Em9) to 36 (Me4-Em3) (Table S2), with an average of 24. The PPB of the 19 PCs varied from 81.82% (Me7-Em14) to 100% (Me2-Em8 and Me5-Em11), with a mean value of 92.37%. The PIC values for each PC ranged from 0.242 (Me4-Em5) to 0.3637 (Me5-Em11), with an average of 0.2983 (Table S2), demonstrating that it was the most informative PC for genetic diversity studies among *E. breviaristatus* genotypes. Two methods were performed to calculate outlier loci among the seven populations of *E. breviaristatus*. In total, Dfdist detected 36 outliers (8.76% of all 411 SRAP loci, Figure 2a). Only one of these was shared (0.24% of all 411 polymorphic loci, Figure 2b) from analyses with BayeScan software when $\log_{10}$ PO > 0.5 (PO, namely, BF, Bayes factors). For this common outlier, we should focus on whether this locus is related to the local adaptability of *E. breviaristatus*.

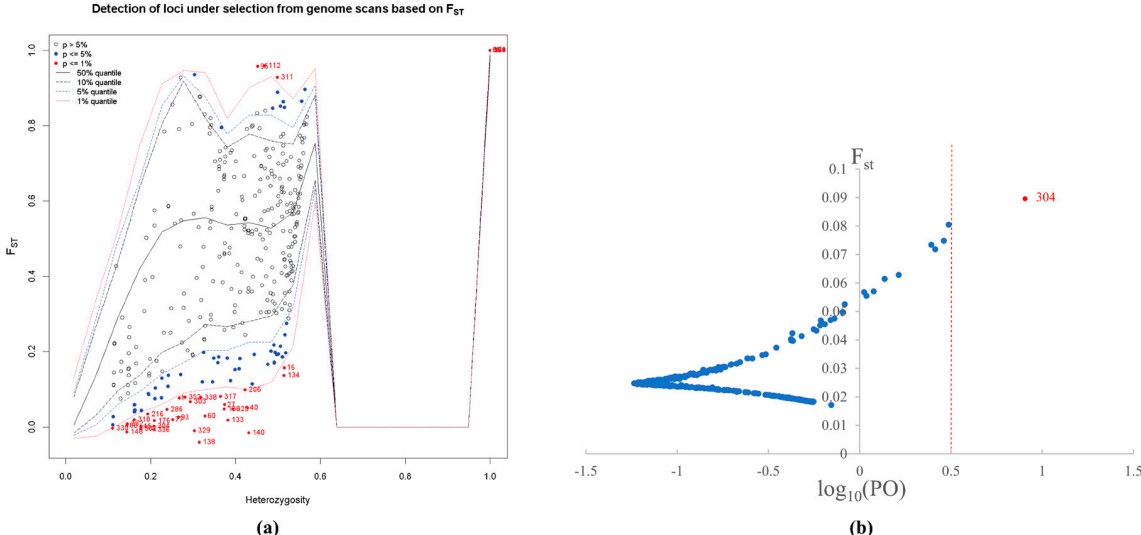

**Figure 2.** Outliers of the seven *E. breviaristatus* populations. The red points are the outlier loci. (**a**) was based on Dfdist. (**b**) was based on BayeScan.

### 3.2. Intra-Population Variation

At the population level, the PPB ranged from 41.18% to 68.63%, with an average of 52.88%. The effective number of alleles (Ne) per locus was 1.3214 (Table 2). The gene diversity ($H_e$) (Nei, 1973) varied from 0.1491 to 0.2494, with an average of 0.1859. Shannon's information index (I) ranged from 0.1790 to 0.3198, with an average of 0.2406, expected Bayesian heterozygosity ($H_B$) varied from 0.1932 to 0.2698, with an average of 0.2224. The expected heterozygosity ($H_j$, analogous to $H_e$) ranged from 0.1718 to 0.2715, with an average of 0.2086. The values of $H_e$, I, $H_B$, and $H_j$ showed a similar trend with PPB (Table 2). Among the seven populations investigated, Pop4 showed the highest genetic variation ($H_e$ = 0.2494; I = 0.3198; PPB = 68.63%), whereas the lowest variation was found in Pop6 ($H_e$ = 0.1491; I = 0.1790; PPB = 41.18%). At the species level, the $H_e$, I, $H_B$, and $H_j$ values equaled 0.3314, 0.4490, 0.3608, and 0.3812, respectively. Levels of pairwise LD for each population ranged from 0.4614% to 1.8824%, while there were 20.37% of LD in all of the 103 *E. breviaristatus*, with the false discovery rate of 0.05. This result indicates that there was lower recombination and replacement in each pop than species. The results in Table 2, as well as the fact that all individuals analyzed were not identical, indicate that *E. breviaristatus* has a considerable degree of genetic variation.

### 3.3. Genetic Differentiation among Populations

The coefficient of genetic differentiation among populations ($G_{ST}$) was 0.4393, as estimated by the partitioning of the total Nei's gene diversity (Table 3), indicating that genetic variation among populations and within population accounted for 43.93% and 56.07% of the total variation, respectively. Shannon's index partitioned 46.41% of the total variation among populations. The genetic differentiation among populations ($F_{st}$) was 0.3478, and the measure of the ratio of genetic diversity among populations ($G_{ST}$-B) based on a Bayesian method was 0.3836, which is consistent with other results. Furthermore, the level of gene flow (Nm) among different populations was low (0.6381 individuals per generation; Table 3). The AMOVA of the total variation showed that 51.1% ($p < 0.001$) was due to variation within populations, whereas the remaining 48.9% was explained by variation among populations (Table 4). Thus, AMOVA ($\Phi_{ST}$ = 0.489) also supported the population divergence from Nei's gene diversity statistics and Shannon's information measure.

**Table 2.** Genetic diversity indexes of seven *E. breviaristatus* populations.

| Population | Sample Size | No. of PB | PPB (%) | $A_o$ | $A_e$ | $H_e$ | $H_o$ | $H_B$ | $H_j$ | LD (%) |
|---|---|---|---|---|---|---|---|---|---|---|
| Pop1 | 13 | 289 | 62.96 | 1.6296 ± 0.4834 | 1.3939 ± 0.3789 | 0.2283 ± 0.2026 | 0.3070 ± 0.2667 | 0.2571 ± 0.0058 | 0.2532 ± 0.0095 | 1.8824 |
| Pop2 | 15 | 215 | 46.84 | 1.4684 ± 0.4995 | 1.2646 ± 0.3593 | 0.1544 ± 0.1954 | 0.1968 ± 0.2378 | 0.1962 ± 0.0070 | 0.1770 ± 0.0092 | 0.526 |
| Pop3 | 15 | 267 | 58.17 | 1.5817 ± 0.4938 | 1.3582 ± 0.3857 | 0.2061 ± 0.2055 | 0.2721 ± 0.2627 | 0.2407 ± 0.0062 | 0.2286 ± 0.0096 | 0.6967 |
| Pop4 | 15 | 315 | 68.63 | 1.6863 ± 0.4645 | 1.4359 ± 0.3875 | 0.2494 ± 0.2035 | 0.3198 ± 0.2576 | 0.2698 ± 0.0059 | 0.2715 ± 0.0096 | 0.812 |
| Pop5 | 15 | 204 | 44.44 | 1.4444 ± 0.4974 | 1.2708 ± 0.3668 | 0.1566 ± 0.1981 | 0.2060 ± 0.2528 | 0.2029 ± 0.0067 | 0.1796 ± 0.0093 | 0.9181 |
| Pop6 | 15 | 189 | 41.18 | 1.4118 ± 0.4927 | 1.2594 ± 0.3645 | 0.1491 ± 0.1986 | 0.1790 ± 0.2380 | 0.1932 ± 0.0073 | 0.1718 ± 0.0093 | 0.4614 |
| Pop7 | 15 | 220 | 47.93 | 1.4793 ± 0.5001 | 1.2668 ± 0.3515 | 0.1577 ± 0.1929 | 0.2036 ± 0.2389 | 0.1968 ± 0.0066 | 0.1788 ± 0.0091 | 1.2134 |
| Means | 14.7 | 242.7 | 52.88 | 1.5288 ± 0.1038 | 1.3214 ± 0.0734 | 0.1859 ± 0.0413 | 0.2406 ± 0.0577 | 0.2224 ± 0.0050 | 0.2086 ± 0.0094 | 0.93 |
| Species | 103 | 424 | 92.37 | 1.9237 ± 0.2657 | 1.5723 ± 0.3306 | 0.3314 ± 0.1580 | 0.4490 ± 0.2238 | 0.3608 ± 0.0031 | 0.3812 ± 0.0092 | 20.3691 |

Notes: PB, polymorphic bands; PPB, percentage of polymorphic bands; $A_o$, number of alleles per locus; $A_e$, effective number of alleles per locus; $H_e$, Nei's genetic diversity (assuming the Hardy-Weinberg equilibrium); $H_B$, expected Bayesian heterozygosity (without assuming the Hardy-Weinberg equilibrium); $H_j$ (analogous to $H_e$), genetic diversity by AFLPsurv; $H_o$, Shannon's information index.

**Table 3.** Genetic differentiation within and among populations of *E. breviaristatus*.

| Nei's Gene Diversity | | Shannon's Information Index | | Lynch & Milligan (AFLPsurv) | | Bayesian Method | |
|---|---|---|---|---|---|---|---|
| $H_T$ | 0.3314 | $H_{sp}$ | 0.4490 | Ht | 0.3812 | $H_T$-B | 0.3608 |
| $H_S$ | 0.1859 | $H_{pop}$ | 0.2406 | Hw | 0.2486 | $H_S$-B | 0.2224 |
| $H_S/H_T$ | 0.5607 | $H_{pop}/H_{sp}$ | 0.5359 | Hw/Ht | 0.6522 | $H_S$-B/$H_T$-B | 0.6164 |
| $G_{ST}$ | 0.4393 | $(H_{sp}-H_{pop})/H_{sp}$ | 0.4641 | Fst | 0.3478 | $G_{ST}$-B | 0.3836 |
| Nm | 0.6381 | – | – | – | – | – | – |

Notes: $H_S$, $H_W$, Hpop and $H_S$-B, within population gene diversity; $H_T$, Hsp, Ht and $H_T$-B, total gene diversity; $H_S/H_T$, Hpop/Hsp, $H_S$-B/$H_T$-B and Hw/Ht, ratio of gene diversity within population; $G_{ST}$, genetic differentiation coefficient; Nm, gene flow estimated from $G_{ST}$; (Hsp-Hpop)/Hsp, $G_{ST}$-B and $F_{st}$, ratio of genetic diversity among population.

**Table 4.** Hierarchical partitioning of genetic variance using AMOVA.

| Different Pops | Source of Variation | df | Sum of Squares | Variance Components | Percentage of Variation (%) |
|---|---|---|---|---|---|
| Total | Among Pops | 6 | 3402.308 | 567.0514 | 48.90% |
| | Within Pops | 96 | 3610.41 | 37.60844 | 51.10% |
| | Total | 102 | 7012.718 | | |
| Regions | Among Regions | 1 | 795.915 | 6.276007 | 8.15% |
| | Among populations within regions | 5 | 2608.055 | 32.97922 | 42.82% |
| | Within populations | 96 | 3625.467 | 37.76528 | 49.03% |
| | Total | 102 | 7029.437 | | |
| Altitude | Among groups | 2 | 964.4179 | 482.2089 | 0.00% |
| | Among populations within groups | 4 | 2438.802 | 609.7004 | 50.78% |
| | Within populations | 96 | 3613.344 | 37.639 | 49.22% |
| | Total | 102 | 7016.563 | | |
| AMT | Among groups | 2 | 959.8829 | 479.9415 | 0.00% |
| | Among populations within groups | 4 | 2443.337 | 610.8341 | 50.83% |
| | Within populations | 96 | 3613.344 | 37.639 | 49.17% |
| | Total | 102 | 7016.563 | | |
| AMP | Among groups | 2 | 1191.462 | 595.7312 | 1.83% |
| | Among populations within groups | 4 | 2211.757 | 552.9393 | 47.29% |
| | Within populations | 96 | 3613.344 | 37.639 | 50.89% |
| | Total | 102 | 7016.563 | | |

Notes: Two AMOVAs including nested analysis (between regions, among populations within regions and within populations; between groups, among populations within groups and within populations) and among population analysis (among populations and within populations) and were used. Statistics include df = degrees of freedom; SS = sum of squares; VC = variance components estimates; %Total = percentage of total variance contributed by each component. $p < 0.001$ (significance test after 9999 permutations).

The seven populations could be grouped into two regional groups based on collection site, and differentiation between regions was small, explaining 8.15% of the variance, with 42.82% variance explained among populations within regions. Different eco-geographic factors divided the populations into three groups. Based on the altitude of each pops, three groups were divided with a height of 300 m. There was no differentiation among groups, and the 50.78% of the total variability among populations

within groups can be explained by altitude. For AMT, every size of 55 mm precipitation was divided into a group, seven populations were divided into three groups. The 50.83% of the total variance can be explained among populations within groups, with no differentiation among groups. The populations were grouped according to the AMP of sampling sites, and the interval size was 2 °C, which divided populations into three groups. The variability among the defined groups was not significant, with only 1.83% of the variance that could be explained among groups.

### 3.4. Inter-Population Structure

Both Nei's (1978) unbiased genetic distance (D) and genetic identity (I) were calculated for paired comparisons of the seven populations (Table S3). The mean value of D was 0.225. The highest genetic distance (D = 0.292) among the seven populations was found between Pop3 and Pop7 (Table S3). The lowest genetic distance (D = 0.1105) was observed between Pop3 and Pop4 (Table S3). In order to illuminate the relationships between populations and individuals, clustering analysis based on the UPGMA method was used to generate two phenograms (Figures 3 and 4). On the whole, the individuals belonging to the same populations were grouped together. These major clades were all supported by a high bootstrap value (Figure 3).

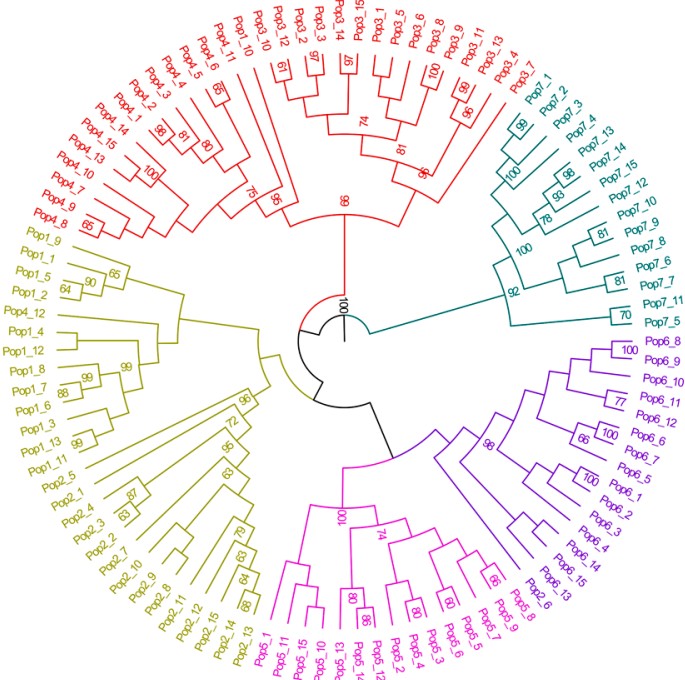

**Figure 3.** UPGMA dendrogram based on individual pairwise Dice similarity coefficients showing relationships among 103 individuals of *E. breviaristatus*. For population designations, refer to Table 1.

The optimal number of genetic clusters (K) was 5, as Ln K value decreased progressively as K was increased from 5 to 7 (Figure 5a), and the highest modal value of ΔK was also at K = 5 (K value ranged from 2 to 6) (Figure 5b). Thus, STRUCTURE divided all individuals into five distinct groups (Figure 4), and the population distribution was consistent with the UPGMA tree (Figure 4). Heterogeneity or intermediacy among populations was reflected in the proportional distribution of individuals. A few individuals shared part of their genome with other populations. Particularly in Pop1, Pop5, and Pop7, many individuals were admixed, sharing parts of their genome with other populations due to common ancestry or gene flow. In addition, based on pairwise Dice similarities, the individual-based PCoA analysis revealed that the first three components explained 22.4%, 19%, and 13.5% of the total variation (Figure 6), supporting the results of STRUCTURE and UPGMA dendrogram of populations

(Figure 4). The PCoA clearly classified all individuals into five groups, with almost all individuals from a population grouped together.

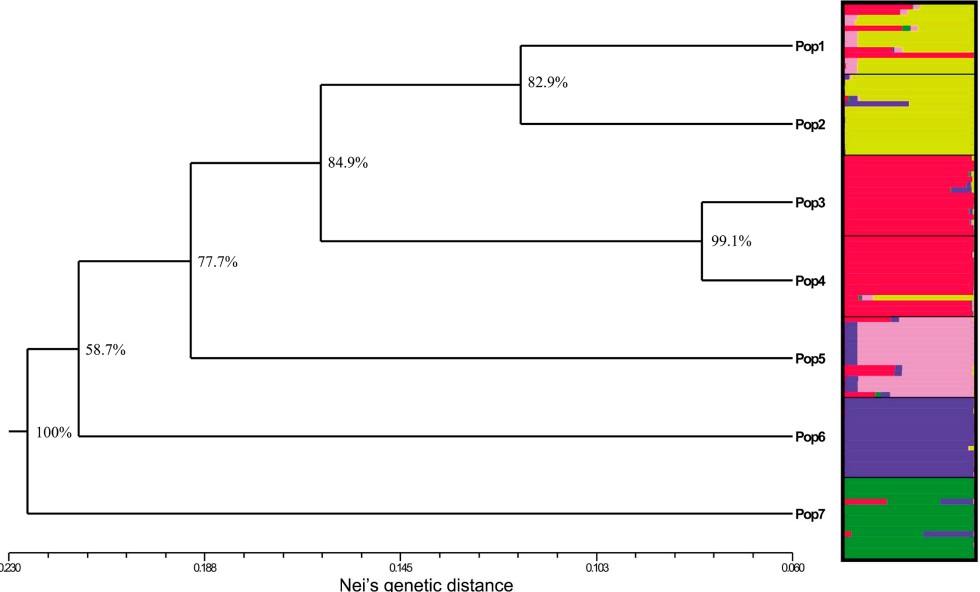

**Figure 4.** The UPGMA dendrogram on the left, based on Nei's genetic distance showing relationships among seven populations of *E. breviaristatus*. The figure on the right estimated population structure of *E. breviaristatus* for a K = 5 population model, which inferred by a Markov's chain Monte Carlo Bayesian clustering method (STRUCTURE 2.3). Each individual is represented by a vertical line, which is partitioned into a maximum of K = 5 differently colored segments that represent the individual's estimated membership fractions in five clusters. Vertical black lines separate individuals of the seven different populations. Populations are labeled below the figure with the abbreviations used in Table 1.

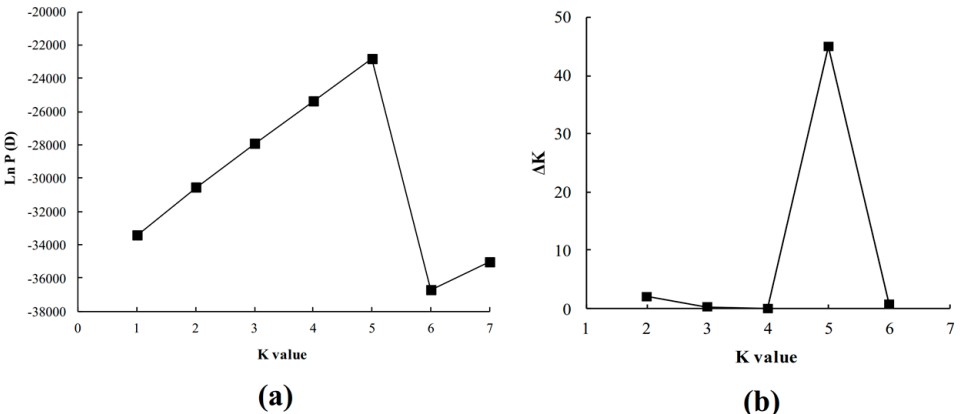

**Figure 5.** Structure analysis of sequence-related amplified polymorphisms (SRAPs) data for the *E. breviaristatus*: (**a**) Mean Ln (K) value (±SD) over ten runs for each K value (K = 1–7); (**b**) values of ΔK [29] are plotted over ten runs for each K value (K = 2–6).

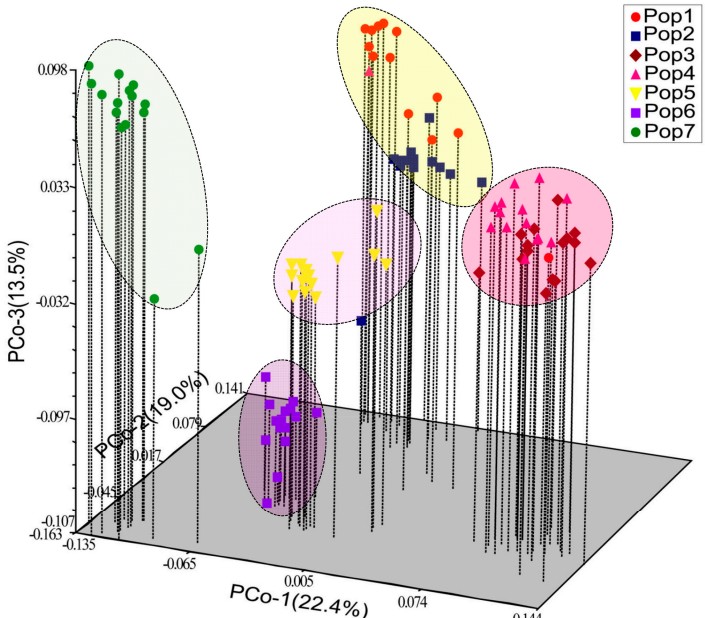

**Figure 6.** Plots of the first three principal coordinate scores showing patterns of relationships among 103 *E. breviaristatus* individuals estimated from the matrix of individual pairwise Dice similarity coefficients. For population designations, refer to Table 1.

### 3.5. The Identification of Genetic Barriers

Barrier analysis identified two predicted barriers among the seven *E. breviaristatus* populations, in which we considered only those with the number over 50 bootstrap value. The first barrier clearly divided the populations into Cluster I and Cluster II. Pop5 and Pop7 were separated from the remaining *E. breviaristatus* populations and formed Cluster I, with the others forming Cluster II, whereas the second barrier separated Pop6 from Cluster II (Figure 7).

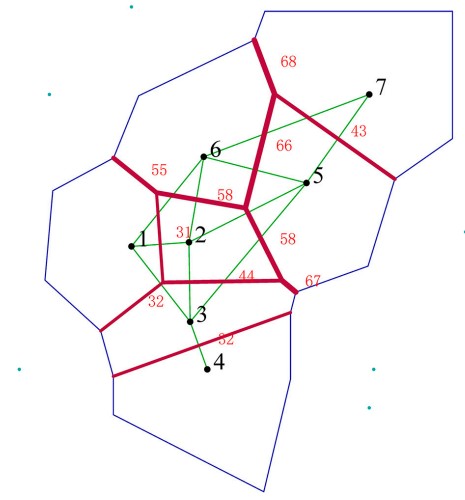

**Figure 7.** Map of genetic barriers detected by BARRIER version 2.2 based on genetic distance. Blue lines show the Voronoï tessellation of populations according to geographic locations, and green lines show the corresponding Delaunay triangulation. Black points represent the locations of seven *E. breviaristatus* populations, and the black number of 1, 2, 3, 4, 5, 6, and 7 represent the populations of Pop1~Pop7, respectively. Red lines with the red numbers represent genetic barriers. The first barrier is represented by the red line with number of 68, 66, 58, and 67. The second barrier is represented by the red line with the numbers 55 and 58.



*3.6. Correlation of Genetic Diversity with Geographic Distances and Eco-Geographic Factors*

The correlation between inter-population Nei's (1978) genetic distances and geographic distances computed by the Mantel test showed a non-significant result (r = 0.2197, *p* = 0.2558), indicating the presence of gene flow or similar genetic structure in the studied populations. The Mantel analysis showed no significant correlation between Nei's (1978) genetic distance and eco-geographic factors (for altitude, r = −0.1269, *p* = 0.2593; for AMT, r = −0.2542, *p* = 0.1155; for AMP, r = −0.0693, *p* = 0.4039). Finally, the Spearman correlation analysis was performed between eco-geographic factors (altitude, AMT, and AMP) and population diversity ($H_e$, $H_o$, $H_B$, and $H_j$) (Tables S4 and S5), but no significant correlations were detected.

## 4. Discussion

*4.1. Genetic Diversity at Species and Population Levels*

Analyzing genetic diversity, assessed by expected heterozygosity ($H_e$) and Shannon's index ($H_o$), can help understand the relationship between the genetic variation's size and space-time distribution and environmental conditions of species thus provide a powerful tool for developing conservation strategies for endemic species [4]. In this study, the SRAP markers revealed moderately high values of genetic diversity in *E. breviaristatus*. When the mean expected heterozygosity ($H_e$ = 0.186) and Shannon's index ($H_o$ = 0.2406) calculated at the population level were compared with other *Elymus* species analyzed with dominant markers, such as RAPD, ISSR, and AFLP (Table S6), *E. breviaristatus* exhibited a relatively low level of diversity, similar to *E. nutans* [21] and *E. trachycaulus* [32]. In contrast, some *Elymus* species, such as *E. alaskanus* [33] and *E. fibrosus* [34], showed even lower estimates of $H_e$ compared with *E. breviaristatus* (Table S6). Additionally, despite its rarity, *E. breviaristatus* showed high species-level diversity that was equivalent to or even higher than that of widespread *Elymus* species such as *E. trachycaulus* [32], *E. caninus* [35], *E. fibrosus* [34], and *E. sibiricus* [36] (Table S6). Similar pattern were also found in many cases [37–39]. However, a meta analysis of 247 plant species indicated that rare species have significantly lower species-level mean population-level measures of variation than common congeneric species [40]. Hence, except geographic range of distribution, there are several other factors that influence the levels of genetic diversity within and among populations, such as ecological and biogeographical factors and mating systems [37–40]. Here, we could give some possible driving factors on genetic variability of *E. breviaristatus*. Firstly, in spite of self-pollination, the gene flow of *E. breviaristatus* could be promoted by the strong wind of QTP, which can augment the $H_{es}$ at the species level and $H_{os}$ at the population level. Secondly, this species is perennial plant and have short rhizomes in growth living habit, whose propagated progenies and daughter ramets helped these current fragmented populations to maintain their historical genetic diversity for a long time when these populations were connected [41]. In addition to molecular markers, allozyme markers are normally also one of the important method for genetic diversity studying though the result were different for *E. breviaristatus* populations from similar eco-region [18]. As dominant markers, SRAP markers could reveal higher genetic polymorphism in *E. breviaristatus* than with allozyme markers (PPB = 92.37% for SRAP markers and PPB = 57.1% for allozyme markers, Table S6), which showed that the SRAP markers had more effectiveness than allozyme in analyzing genetic diversity, which was also proved by Wang et al. [42].

*4.2. Population Genetic Structure*

On the basis of the STRUCTURE analysis, we found that the inferred genomic fraction of *E. breviaristatus* populations was five (K = 5), and the similar result was revealed by UPGMA and PCoA analysis, which was lower than its naturally sampled number (7). This could be manifested by the relatively low genetic distance between Pop1 and Pop2 (GD = 0.1493), and that between Pop3 and Pop4 (GD = 0.1105), with the close geographically distance, which could lead a relatively high gene flow and great genetic similarity. And it is very clear that the three remaining populations were

extremely different from each other. Gene flow is essential for understanding the evolution of plants and the population processes within and among species [42]. Even if there was subtle genetic structure, its function may be weakened by foreign immigrants [43]. Therefore, the higher level of population genetic differentiation among *E. breviaristatus* populations was also subjected to small amounts of gene flow (Nm = 0.6381), which was lower than that of more widespread plants (Nm = 1.881) [44]. This could be explained by the breeding system (reported as self-pollinating, but no analysis of selfing rates exist [17]) and special ecological environments (the geographical distances). A meta-analysis based on RAPD studies by Nybom [44] showed that most genetic variation within populations is retained in long-lived, outcrossing species. In addition, genetic differentiation ($\Phi_{ST}$) of short-lived perennials, endemic, regional, and mixed breeding systems was 0.41, 0.26, 0.42, and 0.40, respectively. The result from our study ($\Phi_{ST}$ = 0.4890) is consistent with these values, suggesting that *E. breviaristatus* is a short-lived perennial, is endemic, and has a low outcrossing rate and significant genetic heterogeneity.

Genetic differentiation is an important indicator of the genetic structure of a species [45]. Hamrick et al. [46] and Nevo [47] demonstrated that there is a non-random distribution of genetic variation in populations and species. In this study, 43.93% of the variation was differentiated among populations of *E. breviaristatus* (Table 3). This result was also confirmed by AMOVA, which indicated that the high level of molecular variance was due to diversity among populations (Table 4). According to allozyme studies, most genetic variation was observed among populations in self-pollinated species, whereas the opposite is true in outcrossing species [48]. Similar results have been reported in *E. glaucus* ($G_{ST}$ = 0.401) [49], *E. sibiricus* ($G_{ST}$ = 0.425) [36], and *E. nutans* ($G_{ST}$ = 0.4297) [43].

In addition, high levels of inter-population genetic variation of *E. breviaristatus* may be caused by other factors. First, the geographic distribution of *E. breviaristatus* is extremely narrow [50]. It is distributed in alpine meadows and among shrubs in the NSP [7]. Second, the complex terrain and the AMP (Table 4) of the QTP may hinder dispersal of pollen and seeds among populations, thus promoting population divergence. Finally, habitat fragmentation and grassland degradation due to human activities can lead to high levels of genetic diversity and moderate population variation [13].

### 4.3. Contribution of Geographic Barriers and Bioclimatic Factors to Genetic Structure

According to BARRIER analysis, two predicted barriers exist among the seven *E. breviaristatus* populations. The first barrier was probably formed by Mt. Daxueshan, and the second was probably formed because of the geographic separation among populations (Figures 1 and 7). The low level of gene flow (Nm = 0.6381, <1.0) reinforced genetic divergence among populations. AMOVA results indicated that the altitude and AMP had little effect on gene flow. Therefore, IBE and IBD might be the main effect limiting gene flow among populations [51], which also was supported by one common outlier locus from the two methods tested.

Non-significant correlations were found between genetic distance and geographical distance or eco-geographic factors. Similar results have been reported in previous studies on *Elymus* species [34,36], except in *E. trachycaulus* [32] and *E. nutans* [52]. The reason for the inconsistency of these results might be that the geographic structure of the species ranges was different. There were no significant relationships between eco-geographic factors (altitude, AMT, and AMP) and population diversity ($H_e$, $H_o$, $H_B$, and $H_j$) (Tables S4 and S5). This further suggests that different sample sizes and eco-geographic locations lead to differences in genetic diversity among studies, so samples should represent the habitats and distribution of the species as much as possible.

### 4.4. Suggestion of Conservation

The results from this study suggest that the effects of human activities and environmental degradation may exacerbate the endangered status of *E. breviaristatus*. We recommend both in situ and ex situ conservation strategies to restrict the decline of *E. breviaristatus*. First, an in situ conservation strategy should be adopted to protect and restore all existing *E. breviaristatus* populations, to insure that its genetic resources and evolution are preserved under natural environments. Population size

is an important consideration for the recovery of endangered and rare species [53]. We recommend that farming operations maintain moderate-only grazing in the region to promote the maintenance of genetic variation of small and fragmented populations through frequent gene flow among populations and to prevent the loss of genetic diversity due to random genetic drift. Grazing animals can promote the spread of seeds and increase gene flow when they are active around the pasture [38,53]. Second, for ex situ conservation, we recommend establishing a germplasm resource pool for this species. A reasonable method of ex situ conservation is to collect and preserve germplasm materials from relatively few individuals so as not to disturb populations.

## 5. Conclusions

In conclusion, this is the first study exploring the genetic diversity and population structure of natural *E. breviaristatus* using SRAP markers. Despite moderate genetic diversity within *E. breviaristatus* populations, there is considerable differentiation among populations, which is consistent with its mating system as well as its rarity in nature, leading to isolation and limited gene flow. The results of this work could be used in further research in *E. breviaristatus*, including conservation and management of germplasm organization, quantitative trait locus gene mapping, and evolutionary research.

**Supplementary Materials:** The following are available online at http://www.mdpi.com/2071-1050/11/20/5863/s1, Table S1: Primer sequences used for SRAP analysis, Table S2: Analysis of SRAP-PCR amplification results, Table S3: The unbiased genetic distance (D) and geographic distances (km) of the seven populations, Table S4: The Correlation coefficient between ecological geographical factors and genetic diversity, Table S5: The *p*-value of the correlation coefficient between ecological geographical factors and genetic diversity, Table S6: Analyses of the genetic diversity at the population level and species level with different markers in variance *Elymus* species. Markers of studies in each species, level of genetic variance and PPB values are given.

**Author Contributions:** X.M. devised this study. Z.D. and J.Y. designed the methods. Q.Y., Y.X. (Yi Xiong) and Y.X. (Yanli Xiong) analyzed the data. Q.Y. and Q.L. wrote and revised the manuscript. W.L. and S.B. revised the manuscript.

**Funding:** This research received no external funding.

**Acknowledgments:** We are very grateful to the Department of Grassland Science, Sichuan Agricultural University for providing us with experimental equipment and venue. Thanks to all authors for their hard work on this manuscript. This work was supported by the National Science Foundation of China (31772661) and Sichuan Key Technology R&D Program (2019YFN0170).

**Conflicts of Interest:** The authors declare no conflicts of interest.

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
