# Peer review of "Genetic Diversity and Population Divergence of a Rare, Endemic Grass (Elymus breviaristatus) in the Southeastern Qinghai-Tibetan Plateau"

_sustainability, doi:10.3390/su11205863_

Round 1

Reviewer 1 Report

The present manuscript reports the investigation of genetic diversity and population Divergence of E. breviaristatus populations distributed in the southeast of Qinghai-Tibetan Plateau 

There are several concerns to address which are below listed:

- The authors analyzed 103 samples representing eight populations of E. breviaristatus, although in the result section only seven populations are discussed. Please clarify.

- The authors reported that the optimal number of genetic clusters (K) was 5, although they analyzed 7 different populations. How do the authors explain this result? They may discuss this point results in the discussion section.

- In Fig.2 the authors may underlie the different clades using some colors. This will help the reliability of the paper.

- The authors should clarify the lamp conditions used in the growth chamber (e.g cool white fluorescent lamps 350–400 μmol m−2 s−1).

- In the material and methods section the authors should be more consistent in the reagents used for the experiment. In line 136 they reported “Plant DNA Kit (Tiangen Biotech., Beijing, China) “. By contrast, at line 139 they only write “Tiangen Biotech., Beijing”.

- The result section appears too fragmented. Please consider the possibility to merge some of the paragraphs.

- The discussion section is too scarce compared with the other part of the manuscript. In addition, the authors did not discuss in deep their results with those found in literature.

Author Response

请看附件

Reviewer 2 Report

The study was well-designed and the results are very interesting either as molecular records or conservation biology item. Sometimes (very rare) during the reading of the manuscript I was interrupted by strange words, so ask the native to check the manuscript.

Author Response

请看附件。

Reviewer 3 Report

This paper explores genetic diversity and the hierarchical structure of seven Elymus breviaristatus populations from Qinghai-Tibetan Plateau, China.
The paper is adequately structured and scientifically sound, supported by a proper list of references and based on a considerable set of analytical and statistical methods.
Even though the studied species has a minimal distribution, the outcomes provided by the authors may help to elucidate prevailing mechanisms of genetic isolation in rare plant species. In this sense, the study may be judged beyond its local interest, and it can contribute more broadly to the planning of conservation policies worldwide.
On the other hand, the potential value of the paper is only partially developed in its present form. My primary concern deals with the writing of the discussion, which sounds like a mere replication of the primary outcomes. Although I concur that some statements summarising the results might enhance the fluency of the paper as a whole, I would like to see something more in the paragraph. For example, even though the authors provided some connections between the studied species vs other taxa of the genus, few interpretations are given to that in terms of biological or biogeographical factors that might be hypothesised behind the observed phenomena. Moreover, the paragraph dealing with the implications for conservation, though it is informative, seems not to be supported by the experimental section of the work.
I am also attaching an annotated copy of the pdf with several remarks.
Overall, the manuscript offers original learning on the topic of plant conservation, but it still needs revision before further consideration for publication.

Author Response

请看附件。

Round 2

Reviewer 1 Report

The manuscript has been improved according to my suggestions. It can be accepted

Congratulations

Author Response

Question 1 The manuscript has been improved according to my suggestions. It can be accepted. Congratulations.

Answer: Thank you.

Reviewer 3 Report

I have now re-examined the revised version of the manuscript entitled "Genetic Diversity and Population Divergence of a Rare, Endemic Grass (Elymus breviaristatus) in the 3 Southeastern Qinghai-Tibetan Plateau". The authors successfully addressed my remarks to the previous version of the paper. In particular, details on populations dimensions have been provided, allowing a more robust evaluation of the results. I also find that the discussion, although it is concise, has mainly been improved, highlighting some potential relationships between genetic diversity and environmental driving factors. Basing on these comments, in my opinion, the manuscript in its present form is now suitable for publication in Sustainability.

Author Response

Question 1 I have now re-examined the revised version of the manuscript entitled "Genetic Diversity and Population Divergence of a Rare, Endemic Grass (Elymus breviaristatus) in the Southeastern Qinghai-Tibetan Plateau". The authors successfully addressed my remarks to the previous version of the paper. In particular, details on populations dimensions have been provided, allowing a more robust evaluation of the results. I also find that the discussion, although it is concise, has mainly been improved, highlighting some potential relationships between genetic diversity and environmental driving factors. Basing on these comments, in my opinion, the manuscript in its present form is now suitable for publication in Sustainability.

Answer: Thank you.